# Platinum Based Nanoparticles Produced by a Pulsed Spark Discharge as a Promising Material for Gas Sensors

**Ivan A. Volkov [1], Nikolay P. Simonenko [2], Alexey A. Efimov [1,*], Tatiana L. Simonenko [2], Ivan S. Vlasov [1], Vladislav I. Borisov [1], Pavel V. Arsenov [1] , Yuri Yu. Lebedinskii [1], Andrey M. Markeev [1], Anna A. Lizunova [1] , Artem S. Mokrushin [2], Elizaveta P. Simonenko [2], Vadim A. Buslov [3], Andrey E. Varfolomeev [4], Zhifu Liu [5], Alexey A. Vasiliev [4] and Victor V. Ivanov [1]**

[1] Moscow Institute of Physics and Technology, Dolgoprudny, 141701 Moscow, Russia; volkov256@yandex.ru (I.A.V.); isvlasov5@yandex.ru (I.S.V.); borisov.vi@mipt.ru (V.I.B.); arsenov-pasha@mail.ru (P.V.A.); lebedinskii.iuiu@mipt.ru (Y.Y.L.); markeev.am@mipt.ru (A.M.M.); lizunova.aa@mipt.ru (A.A.L.); ivanov.vv@mipt.ru (V.V.I.)

[2] Kurnakov Institute of General and Inorganic Chemistry, Russian Academy of Sciences, 119991 Moscow, Russia; n_simonenko@mail.ru (N.P.S.); egorova.offver@gmail.com (T.L.S.); artyom.nano@gmail.com (A.S.M.); ep_simonenko@mail.ru (E.P.S.)

[3] JSC "Scientific Research Institute of Electronic Technology", 394033 Voronezh, Russia; vadbus@mail.ru

[4] NRC "Kurchatov Institute", 123182 Moscow, Russia; varfol_ae@mail.ru (A.E.V.); a-a-vasiliev@yandex.ru (A.A.V.)

[5] Shanghai Institute of Ceramics, Chinese Academy of Sciences, Shanghai 201899, China; liuzf@mail.sic.ac.cn

\* Correspondence: newaldan@gmail.com

**Abstract:** We have applied spark ablation technology for producing nanoparticles from platinum ingots (purity of 99.97 wt. %) as a feed material by using air as a carrier gas. A maximum production rate of about 400 mg/h was achieved with an energy per pulse of 0.5 J and a pulse repetition rate of 250 Hz. The synthesized nanomaterial, composed of an amorphous platinum oxide PtO (83 wt. %) and a crystalline metallic platinum (17 wt. %), was used for formulating functional colloidal ink. Annealing of the deposited ink at 750 °C resulted in the formation of a polycrystalline material comprising 99.7 wt. % of platinum. To demonstrate the possibility of application of the formulated ink in printed electronics, we have patterned conductive lines and microheaters on alumina substrates and 20 μm thick low-temperature co-fired ceramic (LTCC) membranes with the use of aerosol jet printing technology. The power consumption of microheaters fabricated on LTCC membranes was found to be about 140 mW at a temperature of the hot part of 500 °C, thus allowing one to consider these structures as promising micro-hotplates for metal oxide semiconductor (MOS) gas sensors. The catalytic activity of the synthesized nanoparticles was demonstrated by measuring the resistance transients of the non-sintered microheaters upon exposure to 2500 ppm of hydrogen.

**Keywords:** spark ablation technology; platinum-based functional ink; aerosol jet printing; printed gas sensors

## 1. Introduction

Platinum is one of the most studied materials in catalysis today and is widely used for manufacturing catalytic converters [1,2], fuel cells [3,4] and thermocatalytic gas sensors [5–7]. Due to its high chemical stability and high melting point, platinum is used for patterning the read-out electrodes and heaters of metal oxide semiconductor (MOS) gas sensors [8–10], such as by printing colloidal platinum over ceramic membranes such as alumina, LTCC and yttria stabilized zirconia (YSZ) [11,12]. In the case of polymer substrates (e.g., polyimide), colloidal gold can be used for this purpose as well [13].

Chemical methods for the synthesis of platinum nanoparticles are the most frequently used. They are based on the reduction of platinum complexes such as chloroplatinic acid [14], $K_2PtCl_6$ [15], $(NH_4)_2PtCl_6$ [16] or platinum(II) acetylacetonate [17]. One of

the most developed methods for the chemical synthesis of metal nanoparticles is the polyol process, which allows for the production of nanoparticles (e.g., Ag, Au, Pt and Pd) with a tailored size and shape in a size range below 10 nm [18]. In this process, the polyol serves both as a solvent and as a reducing agent. The supported platinum catalysts can be fabricated by using tetraammineplatinum(II) nitrate $Pt(NH_3)_4(NO_3)_2$ as a precursor. In this case, the formation of platinum nanoparticles is carried out by incipient wetness impregnation of a porous support, such as $CeO_2$ or $\gamma$-$Al_2O_3$, with an aqueous solution of this precursor, followed by drying and calcination at a temperature of 500 °C [19]. Tetraammineplatinum(II) chromates were also reported as precursors of Pt-$Cr_2O_3$ composite catalysts, obtained by the thermal decomposition of these complex salts [20]. Since the application of chemical methods implies concerns about biosafety, methods based on the green chemistry approach are getting popular. Currently, researchers are exploring the bacteria-, fungi- and plant-based synthesis of nanoparticles, including the synthesis of platinum nanoparticles [21].

Spark ablation technology offers a cost-effective and a scalable route to nanoparticle generation. It enables the production of powders from any starting (feed) materials with a satisfactory electrical resistivity (<0.2 $\Omega \cdot$cm) [22] and is characterized by a number of advantages, such as chemical purity of the produced powders, the absence of chemical wastes, the ability to use any conductor (pure metals, alloys or doped semiconductors) as a feed material and ease of controlling the size characteristics of the particles. Various designs of a spark discharge generator were developed since its introduction by Schmidt-Ott [23]. Meuller et al. reviewed six designs developed by 2012 [24]. The spark ablation technology was successfully applied for producing various nanomaterials [25], including platinum and platinum oxide nanoparticles [26]. Recently, we applied this technology for producing tin oxide nanoparticles for use as a gas-sensing material [27], which exhibited a reduced sensitivity to water vapors due to the low content of surface hydroxyl groups.

The objective of this work was to investigate the possibility of spark ablation technology to enable high-throughput production of platinum-based nanoparticles possessing a high potential of application in printed electronics. The synthesis conditions allowing one to produce nanomaterial composed of PtO and PtO/Pt nanoparticles from platinum ingots as a feed material at a rate of about 400 mg/h have been found. The functional ink formulated based on this nanomaterial demonstrated relatively high colloidal stability and was used for the patterning of conductive lines and microheaters on alumina substrates and thin LTCC membranes. The catalytic activity of the synthesized nanoparticles was also demonstrated by measuring the thermal responses of the non-sintered microheaters to hydrogen.

## 2. Materials and Methods

### 2.1. Synthesis of Platinum-Based Nanoparticles

Platinum-based nanoparticles were produced by using an in-house developed spark discharge generator comprising a high-voltage source, energy storage unit (capacitor) and the chamber [28]. The chamber used was designed so that the removal of aerosol particles from the spark gap between two electrodes was implemented by supplying a carrier gas through one of the hollow electrodes. In this work, two electrodes were prepared from platinum ingots with a purity of 99.97 wt. % by rolling them into tubes with an inner diameter of about 4 mm. The prepared electrodes were fixed coaxially in the copper terminals of a generator circuit. During synthesis, the gap between the electrodes was kept the same by gradually changing the position of one of them as the material from both electrodes was ablated as a result of a spark discharge. This was accomplished by keeping the maximum voltage drop across the gap at the same level (3 kV). The electrical energy (W) stored in the capacitor, which was then transferred to the circuit during the single pulse of a spark discharge, was estimated to be 0.5 J. The synthesis was carried out in an atmosphere of ambient air filtered through a high-efficiency particulate air (HEPA) capsule filter (TSI Inc., Shoreview, MN, USA). Air was selected as a carrier gas because of the need to produce nanoparticles comprising the platinum oxide phase, which was found

to have a much higher affinity for polar solvents used for ink formulation (see Section 2.2) as compared with metallic platinum. The filtered air was fed through one of the electrodes at a flow rate of 3.5 L/min. To provide the maximum production rate of nanoparticles of about 400 mg/h, a pulse repetition rate of 250 Hz was used. Higher repetition rates caused the content of the solidified liquid droplets of platinum 10–50 μm in diameter (hereinafter referred to as platinum droplets) to exceed 20 wt. % of a total mass of collected powder. In the spark discharge process, the droplets were formed as a result of splashing of the partially molten edges of electrodes by the carrier gas, as proposed in [26,29]. One could estimate the mass (m) of the nanoparticles produced per pulse by dividing the production rate by a repetition rate, thus yielding about 440 ng per pulse. Therefore, the electrical energy consumed per unit mass of nanoparticles could be estimated as a W/m ratio, which was equal to $1.1 \cdot 10^6$ J/g, or 0.31 (kW·h)/g.

## 2.2. Ink Formulation

Basically, a colloidal ink is composed of particles of a given functionality, dispersed in a vehicle representing a solvent with dissolved additives. The formulation of platinum-based ink was carried out while accounting for the following requirements. First, the concentration of platinum should be over 20 wt. % in order to enable the fabrication of conductive patterns on ceramic substrates at a moderate dosage of deposited ink, thus accelerating the production process and, hence, decreasing the degree of contamination of the fabricated patterns with particulate matter. Second, the ink should possess relatively high colloidal stability to ensure sufficient reproducibility of the parameters of the printed features.

The composition of the ink was tailored according to several principles. The most important principle consisted of the need to provide high wettability of a solid phase by a vehicle. If the surface tension between a vehicle and a solid phase is below the critical value, a peptization process takes place, and a thermodynamically stable (lyophilic) colloid is formed [30]. The second principle was to use solvents possessing moderate boiling points (100–200 °C) and low toxicities, which was crucial from the viewpoint of ink application [31,32]. Finally, the viscosity of the ink is better to keep between 1 and 25 mPa·s, while its surface tension should be between 30 and 50 mN/m so that it could be utilized in inkjet and aerosol jet printing technologies [31–34].

In this work, the ink was formulated from the spark discharge-synthesized platinum-based nanoparticles by using a binary solvent composed of water and ethylene glycol. In the course of formulation, the concentrations of the polymer and plasticizer entering into the composition of the vehicle were optimized so as to attain both good quality for the initial composite film (dry residue of the ink) formed after solvent evaporation and a relatively high packing density of the nanoparticles in the heat-treated (binder-free) film. The ink preparation procedure involved ultrasonication, facilitating the separation of nanoparticles from the powder material composed of them. The specific ultrasonic power transferred to the ink was estimated to be 3 W/cm$^3$, and the processing period was 1 h. The use of a water cooling system allowed for keeping the temperature of the ink during ultrasonication below 30 °C. After ultrasonication, the platinum droplets (~20 wt. % of the total mass of the powder) were separated from the colloidal nanoparticles through sedimentation in a gravitational field.

## 2.3. Aerosol Jet Printing

To demonstrate the possibility of the application of formulated ink in printed electronics, we patterned conductive lines and microheaters on alumina substrates and low-temperature co-fired ceramic (LTCC) membranes with the use of aerosol jet printing technology. An AJ 15XE commercial aerosol jet printer (Neotech AMT GmbH, Nuremberg, Germany) utilizing the aerodynamic focusing technique was used for this purpose. In this technology, the ink is deposited onto the substrate in the form of aerosol particles generated in the atomizer. The printing resolution (lower size limit of printed features) is determined by the aerosol stream diameter at the substrate surface, which is controlled by

a ratio between the sheath gas flow and the aerosol flow in the nozzle (focusing ratio) and by a nozzle orifice diameter.

Figure 1 presents a stencil of a microheater designed in AutoCAD software, which was then compiled by the firmware of the aerosol jet printer. This stencil was used to print a series of microheaters on a 20 μm thick LTCC membrane covering a 100 μm thick LTCC wafer comprising holes with a diameter of 4 mm. The stencil was positioned with respect to the hole so that its narrowest part would be located in the central area of the hole in order to minimize the power consumption of the microheater when heated by a current.

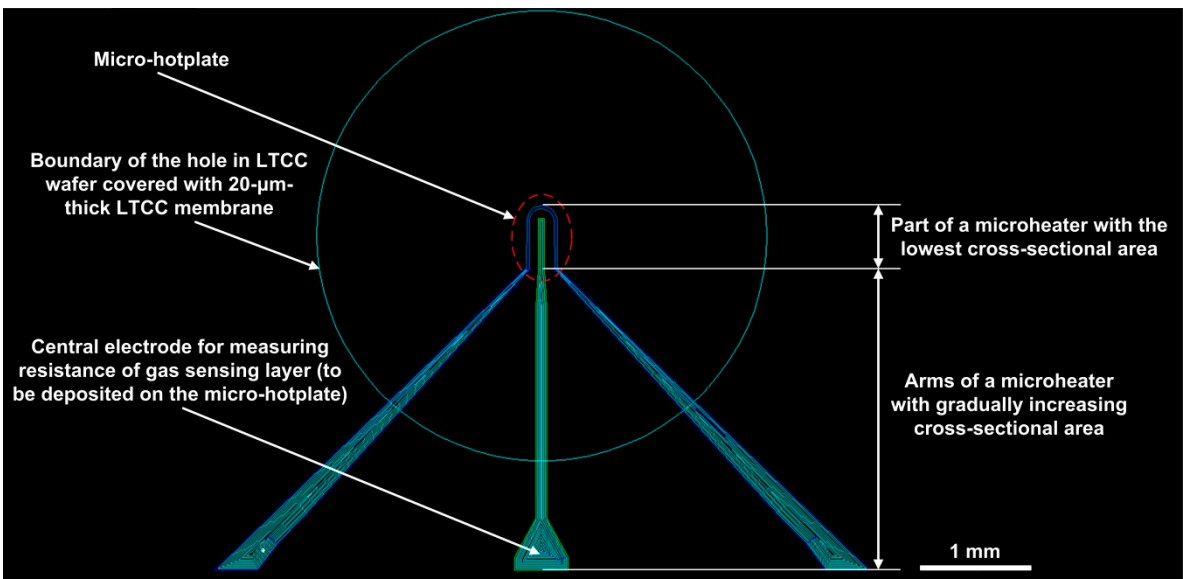

**Figure 1.** Stencil of a microheater used for aerosol jet printing.

The average dosage of deposited ink (mass of the ink per unit area of the covered surface) was tailored so as to provide an average thickness of the printed features in the final (annealed) form in the range of 0.4–0.7 μm, thus exceeding the root mean square (RMS) roughness of the substrates used by a factor of more than 3. When printing the narrowest part of a microheater, the average dosage was controlled by varying the atomizer gas flow, printing speed, substrate temperature and number of consecutive passes of an aerosol jet along a given path. When printing the arms of a microheater, the average dosage was additionally controlled by varying the distance between neighboring paths of the aerosol jet. In the performed experiments, the following operating conditions were used: gas flow in the pneumatic atomizer of 700 sccm, focusing ratio of 2, nozzle orifice diameter of 150 μm, printing speed of 100 mm/min and a substrate temperature of 100 °C.

*2.4. Characterization of Materials*

A number of methods were used to get information about the elemental composition, phase composition, and size characteristics of the powder produced from the feed material by spark discharge and the sintered material obtained through annealing of the deposited ink. The powder samples were taken for analysis from the collected powder so that the majority of the platinum droplets (10–50 μm in diameter) were not brought into the samples. This was possible to execute due to the fact that large objects possessing low specific surfaces tend to be located at the bottom of the powder material. The estimated content of platinum droplets in the samples was less than 2 wt. %. Annealing of the deposited ink was carried out in air atmosphere in the muffle furnace as follows: (1) heating the furnace up to 750 °C at a constant rate of 10 °C/min, and (2) keeping the furnace at 750 °C for 2 h.

The elemental composition was determined by inductively coupled plasma atomic emission spectroscopy (ICP-AES) with the use of a Liberty Series II ICP-OES (Varian

International Inc., Zug, Switzerland). The oxidation state of the platinum was controlled by X-ray photoemission spectroscopy (XPS) with the use of a Theta Probe (Thermo Fisher Scientific, Waltham, MA, USA). The phase composition and crystallinity of the materials were estimated by analyzing X-ray diffraction (XRD) spectra measured with a D8-Advance diffractometer (Bruker, Bremen, Germany), while analysis of the XRD spectra was carried out with the use of the Rietveld refinement method, implemented in X'Pert HighScore Plus software (PANalytical B.V., Almelo, The Netherlands). The particles constituting the powder were characterized by transmission electron microscopy (TEM) with the use of a JEM-2100 (JEOL, Tokyo, Japan). The microstructure and elemental composition of the sintered material were examined by scanning electron microscopy (SEM) combined with energy-dispersive X-ray spectroscopy (EDX) with the use of a JSM-7001F (JEOL, Tokyo, Japan) equipped with a Bruker XLash 6/30 EDX spectrometer. The resistivity of the sintered material was calculated from the resistance of the printed lines by using Pouillet's law. The cross-sectional area of the lines used in the calculations was estimated from the data acquired with an S NEOX non-contact optical 3D profiler (Sensofar, Terrassa, Spain).

In the course of ink formulation, the viscosity of the ink samples was controlled with an SV-10 vibrational viscometer (A&D, Tokyo, Japan). The surface tension was estimated by the pendant drop method with the use of a DSA25S drop shape analyzer (Krüss GmbH, Hamburg, Germany). The concentration of platinum in the ink samples was controlled by Thermogravimetry-Differential Scanning Calorimetry (TGA-DSC) thermal analysis performed on an SDT Q600 (TA Instruments, New Castle, DE, USA). The ink samples were heated at a rate of 10 °C/min in air supplied at a flow rate of 250 mL/min. The size distributions of the colloidal particles were obtained by dynamic light scattering (DLS) with the use of a Zetasizer Nano ZS (Malvern Instruments, Malvern, United Kingdom). The packing density of the nanoparticles in the printed ink after removal of a binder (polymer with plasticizer) as a result of heat treatment at 400 °C was controlled by SEM with the use of a JSM-7001F (JEOL, Tokyo, Japan).

The topography and microstructure of the annealed microheaters (heat treated at 750 °C) were examined by optical profilometry and scanning electron microscopy with the use of an S NEOX (Sensofar, Terrassa, Spain) and an NVision 40 (Carl Zeiss, Oberkochen, Germany). The thermal radiation images of the annealed microheaters were obtained with a FLIR SC655 thermal imaging camera (FLIR Systems, Wilsonville, OR, USA) and then processed in ThermaCAM Researcher Professional 2.10 software. The catalytic activity of the platinum-based nanoparticles was demonstrated by measuring the resistance transients of the non-sintered microheaters (heat treated at 400 °C) upon exposure to 2500 ppm of hydrogen mixed with dry air. A commercial instrument, the Microgas-F (Intera, Moscow, Russia), described in detail in [27], was used in this experiment.

## 3. Results and Discussion

### 3.1. Results of Characterization of Materials

#### 3.1.1. Platinum-Based Powder and Sintered Material

It was found from the ICP-AES analysis that the content of platinum in the powder was 92.7 wt. %, and the dominant impurities were represented by tin (0.077 wt. %), copper (0.074 wt. %), silver (0.033 wt. %), palladium (0.022 wt. %), iron (0.014 wt. %), zinc (0.004 wt. %) and calcium (0.004 wt. %). In the sintered material, the content of platinum was 99.7 wt. %, and the dominant impurities were represented by the same elements: tin (0.086 wt. %), copper (0.072 wt. %), silver (0.037 wt. %), palladium (0.023 wt. %), iron (0.020 wt. %), zinc (0.014 wt. %) and calcium (0.005 wt. %). The observed difference in the content of platinum in these two materials was due to the fact that the particles in the powder were partially oxidized, since air was used as a carrier gas in the synthesis. Recently, platinum oxide nanoparticles were found to form in the course of spark ablation of platinum electrodes when employing air as a carrier gas [26]. It is known that platinum oxides are reduced to platinum ($Pt^0$) at elevated temperatures [35], so the platinum oxide entering into the composition of the inorganic phase of the deposited ink was supposed

to be reduced to a metallic state during the heat treatment and then get sintered, thus forming a polycrystalline material. The hypothesis of the occurrence of platinum oxide in the powder was supported by XPS data. The XPS spectrum of the powder in the Pt4f core level region (Figure 2, upper spectrum) was perfectly fitted by a single doublet comprising a Pt4f$_{7/2}$ line located at a binding energy of about 72.4 eV, which indicated the presence of PtO, according to [35,36]. The XPS spectrum of the sintered material was also fitted by a single doublet (Figure 2, lower spectrum) with a Pt4f$_{7/2}$ line located at about 71 eV, suggesting the complete reduction of platinum oxide to a metallic state [35,36].

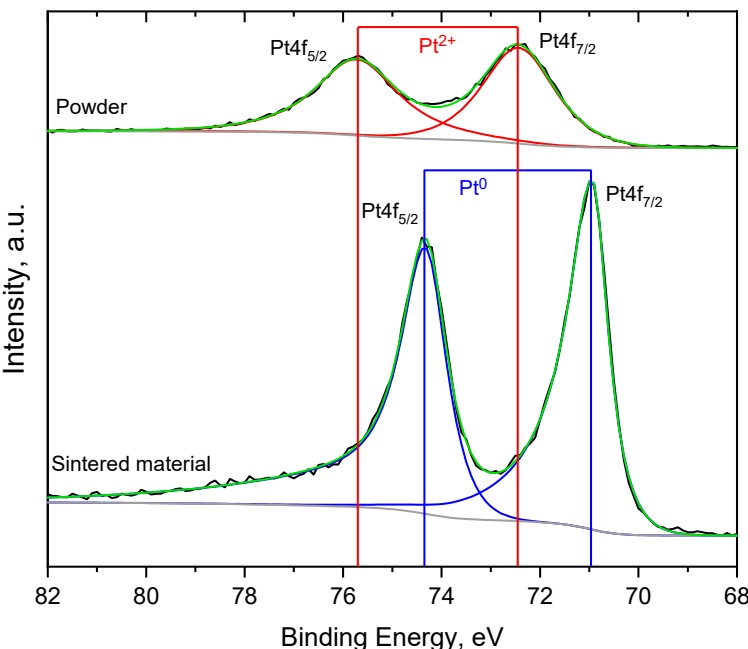

**Figure 2.** X-ray photoemission spectroscopy (XPS) spectra corresponding to the Pt4f core level region of the powder and sintered material. The spectra were acquired by using Al Kα radiation.

It should be noted that the powder had a very limited solubility in the aqua regia used for the preparation of the solution for ICP-AES. Therefore, its dissolution was implemented in two steps. At the first step, the powder was processed in the concentrated hydrochloric acid that led to the dissolution of the platinum oxide. Then, the nitric acid was added to the obtained solution in the required amount in order to dissolve the remaining metallic platinum. At the same time, the sintered material was readily soluble in the aqua regia, as was expected for the metallic platinum.

One may conclude from the analysis of the XRD spectra that both the powder and the sintered material contained only one crystalline phase, represented by platinum with the Fm3m crystal structure (Figure 3A,B). It follows from the XRD spectrum of the powder that there is an amorphous halo in the range of the scattering angles 2θ = 29°–39°, suggesting the presence of an amorphous phase, most likely represented by a platinum oxide since platinum atoms were found to be constituents of the PtO compound, as evidenced by the XPS data (Figure 2). The lines corresponding to platinum atoms involved in the metallic platinum were not observed in the XPS spectrum of the powder because the depth of the XPS analysis is few nanometers, so the photoelectrons emitted from the metallic platinum core of the oxidized particles did not escape from the particles.

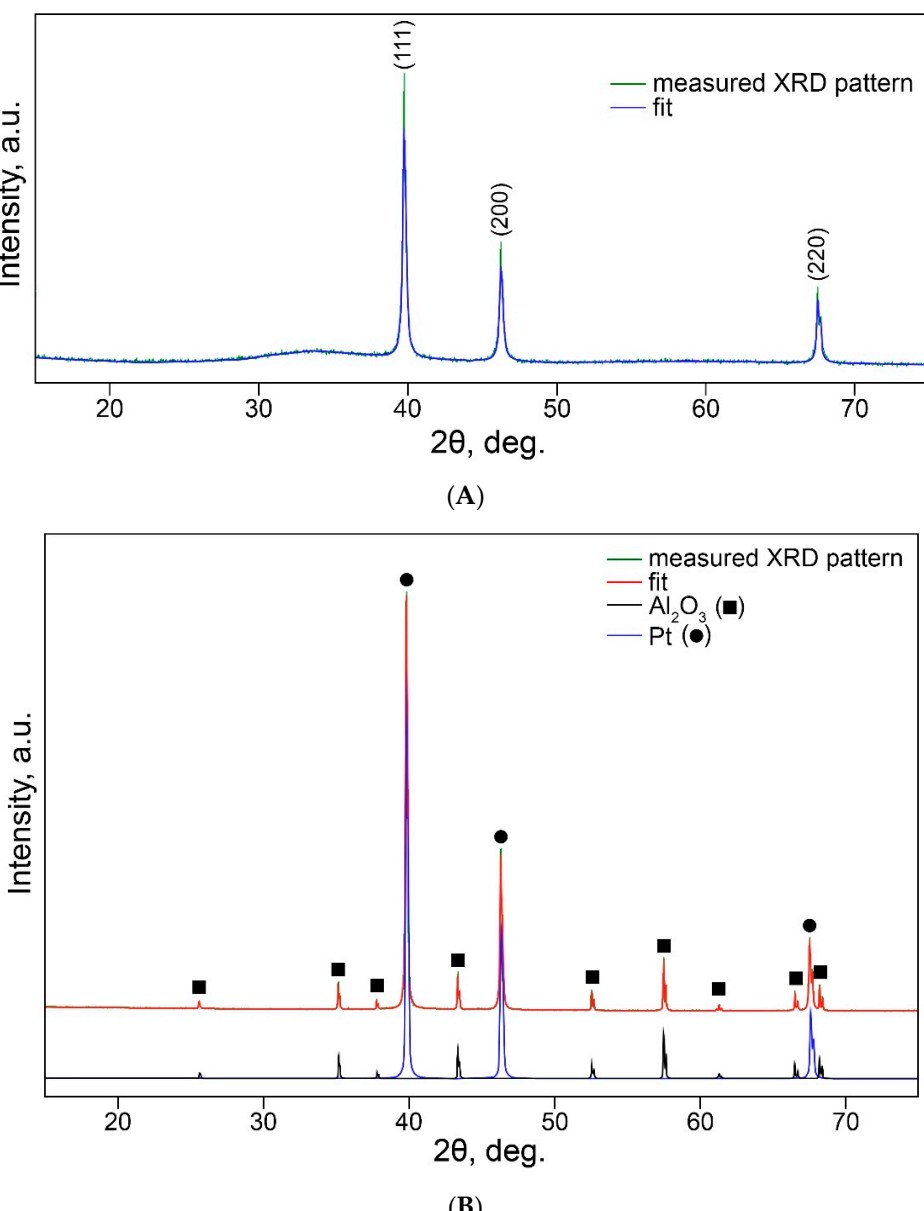

**Figure 3.** *Cont.*

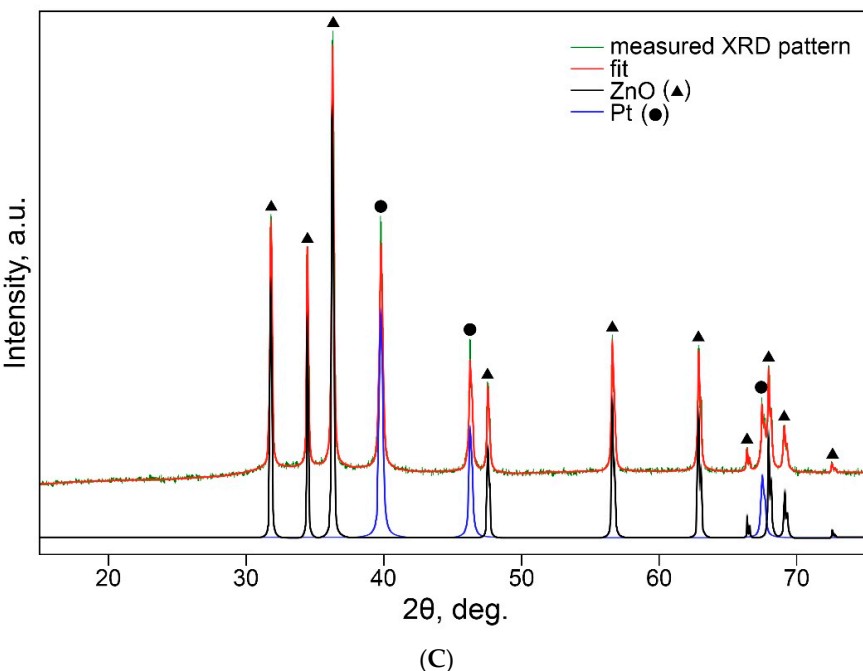

(C)

**Figure 3.** X-ray diffraction (XRD) spectra of the platinum-based powder (**A**), sintered material (**B**) and mixture (1:1 by weight) of the platinum-based powder with ZnO powder, used as a crystalline internal standard (**C**). The spectra were acquired by using Cu Kα radiation. The lines corresponding to $Al_2O_3$ observed in the spectrum of the sintered material are due to the contribution of alumina substrate, used for the deposition of the formulated ink.

To estimate the content of the amorphous phase in the powder, we followed the internal standard method [37]. For this purpose, the platinum-based powder under study was mixed with ZnO powder (Sigma-Aldrich, Saint Louis, MO, USA, no. 544906), used as a crystalline internal standard, at a ratio of 1:1 by weight. Figure 3C presents the measured and calculated patterns of the prepared mixture and the crystalline phases employed in the refinement. The refinement procedure resulted in 14.5 wt. % of the metallic platinum phase and 85.5 wt. % of ZnO in the overall mass of the crystalline phases contained in the mixture of the two powders. Therefore, the content of the metallic platinum phase in the platinum-based powder could be calculated as (14.5/85.5)·100%, thus yielding about 17.0 wt. %. The remaining 83.0 wt. % was accounted for as the amorphous phase (PtO, according to XPS data). The obtained data correlate well with those reported in [26]: 14.4 wt. % of metallic platinum and 85.6 wt. % of the amorphous phase ($PtO_x$, with x = 1.12, as the authors estimated from EDX analysis).

It follows from the TEM images (Figure 4A–C) that the powder was composed of nearly spherical particles, particles of irregular shapes and faceted particles. Some of them exhibited distinct core–shell structures (Figure 4C,D). The selected area electron diffraction (SAED) pattern presented in the inset of Figure 4A consists of rings corresponding to interplanar spacings of 2.25, 1.94, 1.36 and 1.17 Å. These values are in good agreement with those characteristic of (111), (200), (220), and (311) planes of platinum with Fm3m crystal structures. Therefore, the results obtained from the SAED and XRD studies were consistent. Furthermore, taking into account the results of the XPS and XRD examinations, one may conclude that a certain fraction of the powder was represented by amorphous platinum oxide particles, while the other one was represented by particles possessing core–shell structures with a crystalline platinum core and an amorphous platinum oxide shell. Figure 5A demonstrates a histogram representing the size distribution of a number fraction of individual particles, fitted by a log-normal distribution. The calculated, normalized number-weighted and volume-weighted probability density functions are presented in Figure 5B.

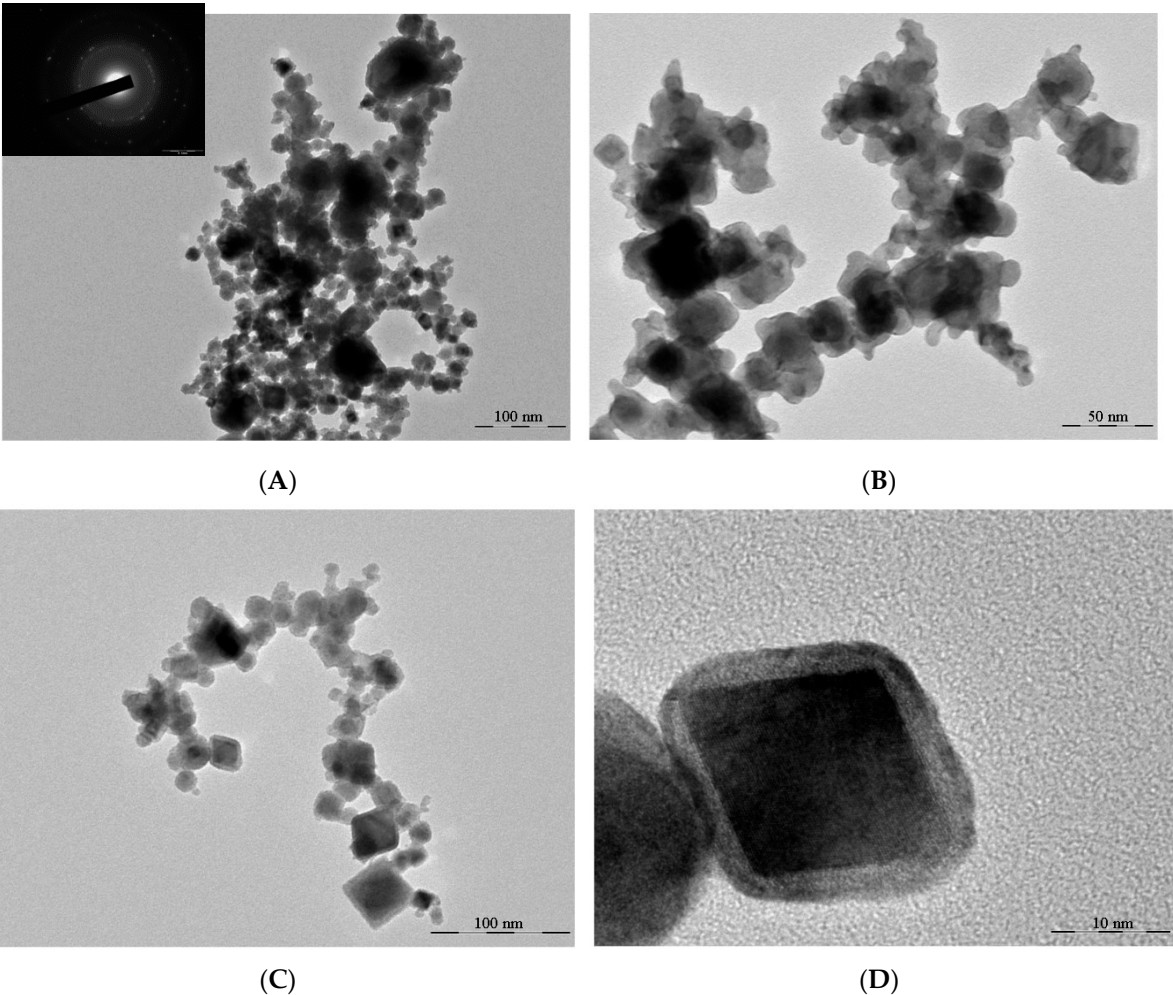

**Figure 4.** Transmission electron microscopy (TEM) images of nanoparticles at different magnifications (**A**–**D**) and the electron diffraction pattern (inset of (**A**)).

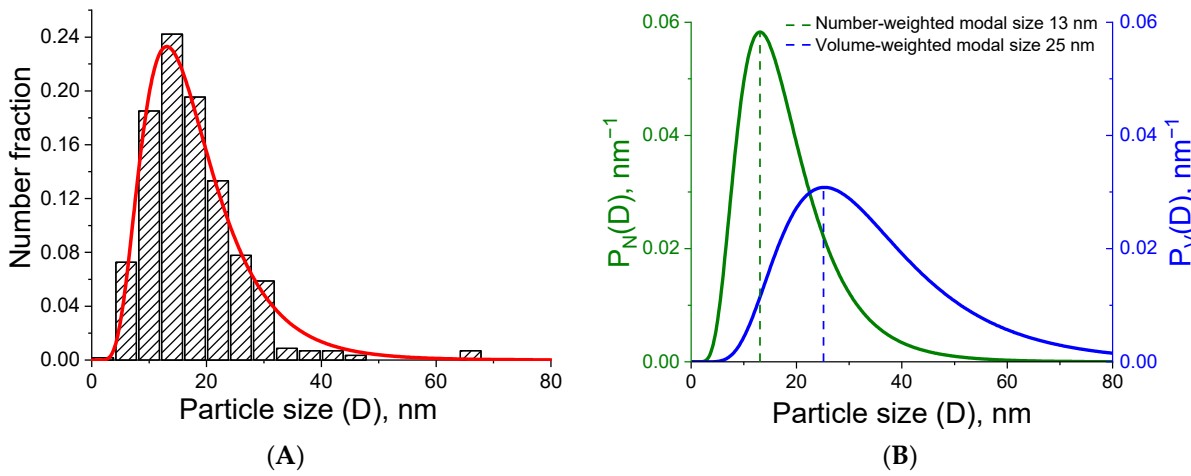

**Figure 5.** Histogram representing the size distribution of a number fraction of individual particles fitted by a log-normal distribution (**A**) and the normalized number-weighted $P_N(D)$ and volume-weighted $P_V(D)$ probability density functions (**B**).

Figure 6 presents the SEM image of the sintered line, printed on alumina substrate, together with the EDX elemental mappings. A typical EDX spectrum of the sintered

material is shown in Figure 7. Figure 8 presents the SEM images of the sintered material at higher magnifications. It follows from these images that the obtained material was characterized by a high degree of sintering. The size of most crystallites was in the range between 100 and 400 nm.

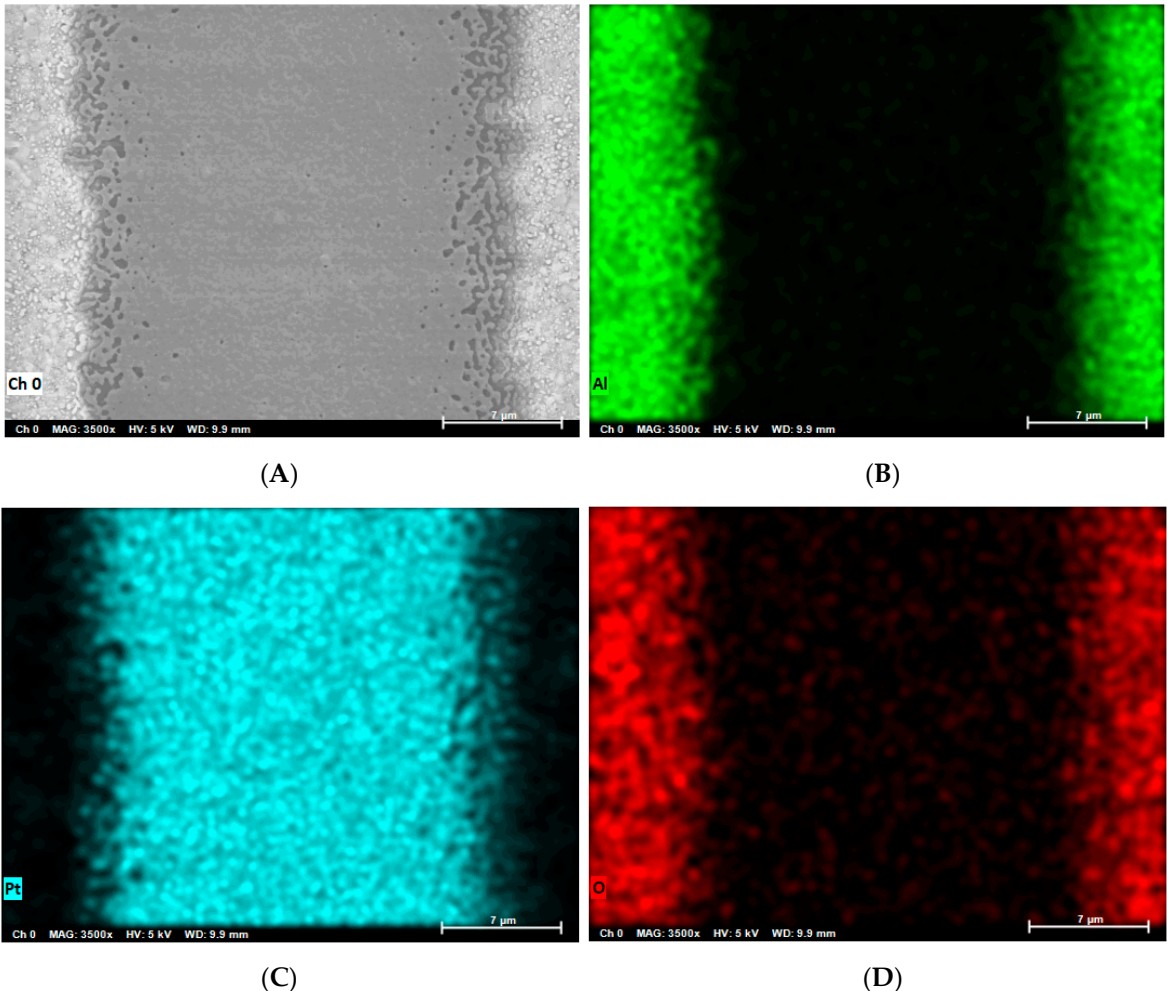

**Figure 6.** Scanning electron microscopy (SEM) image of the sintered line (dark grey central area) printed on alumina substrate (light grey lateral areas) (**A**) and the elemental mappings derived from the following EDX spectra: aluminum (**B**), platinum (**C**) and oxygen (**D**).

The resistivity of the sintered material, calculated based on the results of the measurements of 10 printed lines, was found to be $(1.2 \pm 0.1) \cdot 10^{-7}$ $\Omega \cdot$m at 25 °C, thus exceeding that of bulk platinum by a factor of about 1.1. Since the content of platinum in the sintered material was 99.7 wt. % and the oxidation state of the platinum therein was zero, as evidenced by the ICP-AES and XPS data, the observed difference in the resistivity was due to the insignificant residual porosity of the sintered material (Figure 8).

The results of the characterization of the materials studied are summarized in Table A1, Appendix A.

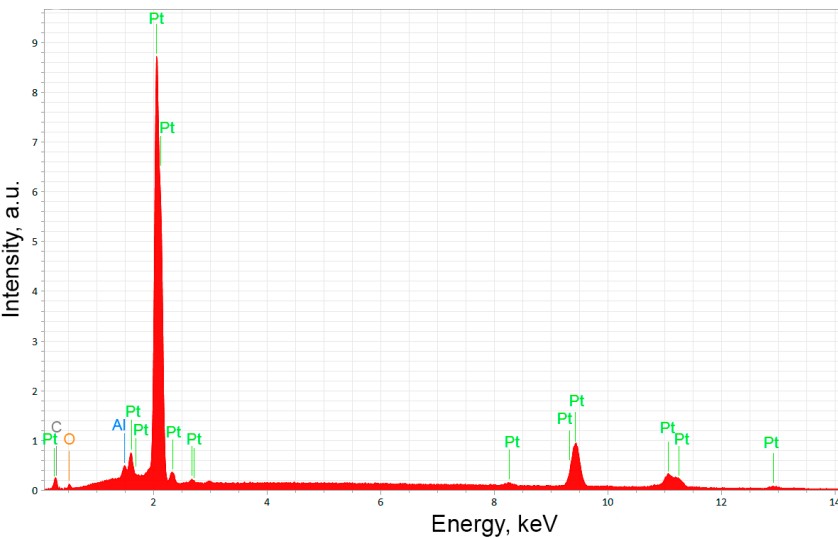

**Figure 7.** Typical energy-dispersive X-ray spectroscopy (EDX) spectrum of the sintered material averaged over the area of $5 \times 5$ μm$^2$. The aluminum and oxygen lines were due to the contribution of the substrate.

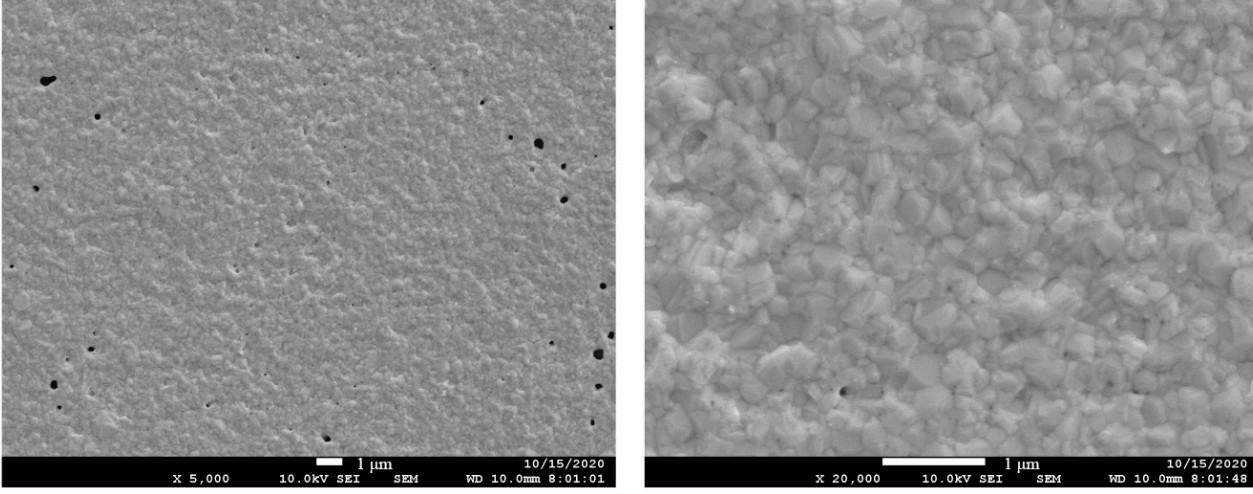

**Figure 8.** SEM images of the sintered line, demonstrating the degree of sintering and the polycrystalline structure of the end material.

### 3.1.2. Platinum-Based Functional Ink

The boiling point of the solvent of formulated ink was measured at an air pressure of 750 mm Hg. The obtained value was equal to 119.6 °C. Figure 9 presents the results of the TG-DSC thermal analysis of the as-prepared ink. In the temperature range up to 170 °C, the loss of the ink mass was mostly due to the evaporation of the solvent components (water and ethylene glycol). A further increase in temperature resulted in the evaporation and oxidation of the plasticizer and the polymer, constituting the binder of the dry residue of the ink. The total mass loss at 750 °C was equal to 75.5 wt. %, so the residual 24.5 wt. % was accounted for by the platinum.

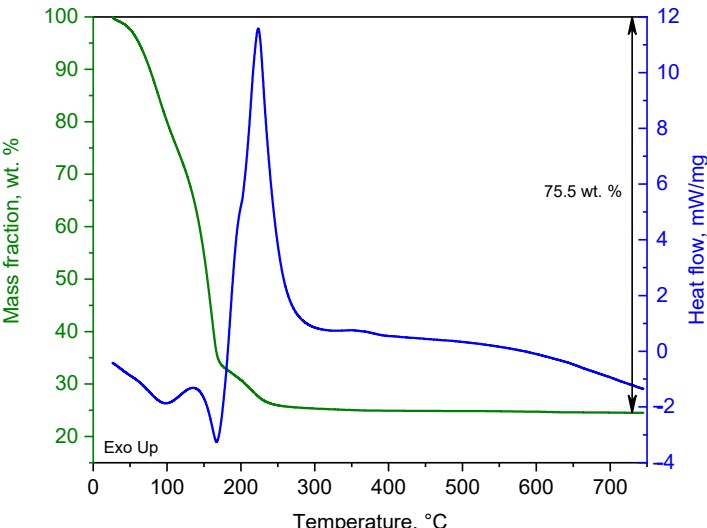

**Figure 9.** Results of the TG-DSC thermal analysis of the as-prepared ink. The green curve is the mass of the sample, and the blue curve is the heat flow per unit mass of the sample. The sample was heated at a rate of 10 °C/min in air supplied at a flow rate of 250 mL/min.

Figure 10 demonstrates the size distributions of the colloidal particles contained in two ink samples. The first one (green curve) is characteristic of the as-prepared ink, and the second one (red curve) is characteristic of the top layer of the ink after 5 days of its storage in a standard 2 mL Eppendorf safe-lock tube kept in a vertical position. The difference in the mean particle size characteristic of these samples, which is a measure of the sedimentation stability of the ink, was about 30%. According to the thermal analysis data, the concentration of platinum in the top layer of the ink after 5 days of storage decreased to 18.4 wt. % as a result of the evolution of the colloid toward the sedimentation–diffusion equilibrium in the gravitational field [30]. Further observations up to 15 days did not reveal changes both in the mean particle size and the concentration of platinum in the top layer within the measurement's accuracy. Gentle agitation of the aged ink (i.e., ink that was stored for several days with no movement) for tens of seconds was sufficient to bring it back to its initial (as-prepared) state. The surface tension of the agitated ink measured at 25 °C was equal to 43.9 mN/m. The viscosity of the ink was 11.4 mPa·s at 25 °C.

Figure 11 presents the SEM images of the nanoparticles contained in the ink. The film under study was formed by annealing the printed ink in ambient air as follows: (1) heating up to 400 °C at a constant rate of 10 °C/min and (2) keeping 400 °C for 2 h. As a result of the heat treatment, the binder (polymer and plasticizer) was removed from the printed ink, and the obtained film represented an array of nanoparticles. It follows from Figure 11 that the film was characterized by a relatively high packing density of nanoparticles, which was important for attaining a high degree of sintering of the end material formed after the annealing at 750 °C.

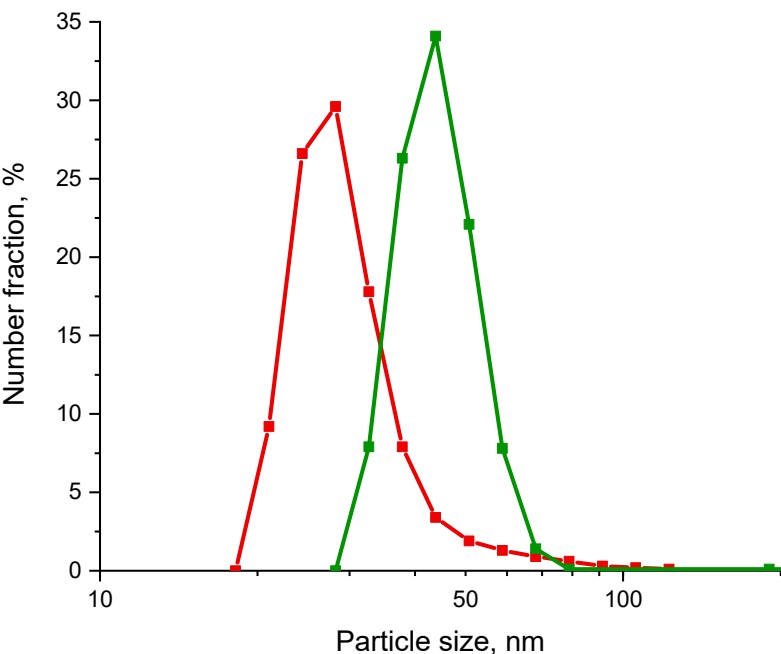

**Figure 10.** Size distributions of the colloidal particles obtained by dynamic light scattering. The green curve is characteristic of the as-prepared ink, and the red curve is characteristic of the top layer of the ink after 5 days of storage.

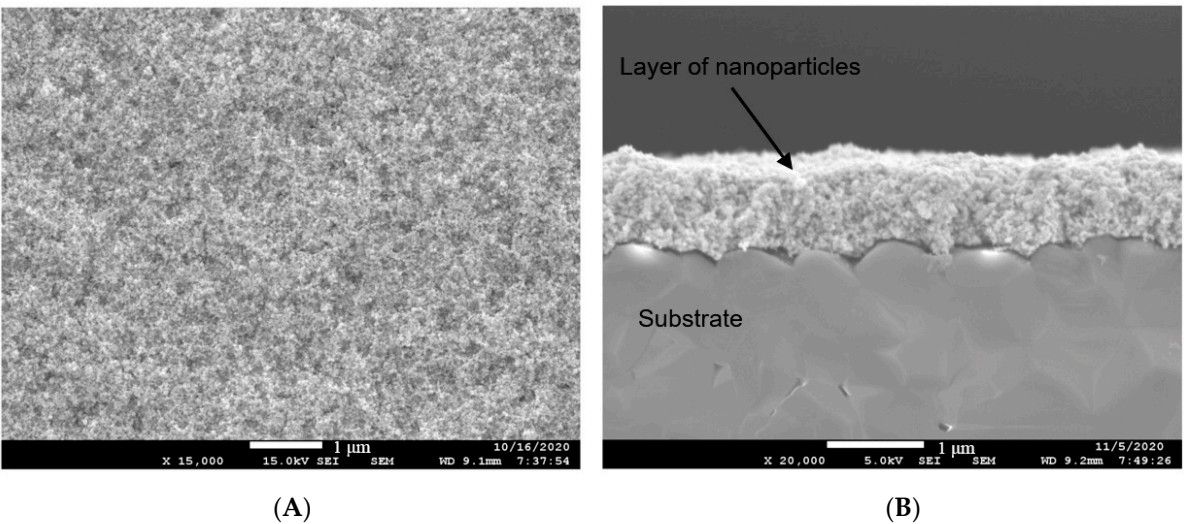

(**A**)                                                                 (**B**)

**Figure 11.** SEM images of the nanoparticles contained in the ink. (**A**) Front surface of the film. (**B**) End surface of the film. The ink was printed on alumina substrate and then annealed in air atmosphere at 400 °C for 2 h to remove the binder.

### 3.2. Topography and Performance of Microheaters

Figure 12 presents images of a typical microheater annealed at 750 °C, which were rendered based on data acquired with a non-contact optical 3D profiler. SEM images are presented in Figure 13. The average width of the narrowest part of the microheater could be estimated to be about 35 µm. The cross-sectional areas of the narrowest part and the initial part of the arms adjacent to the narrowest part are in the ratio of about 1:2.5.

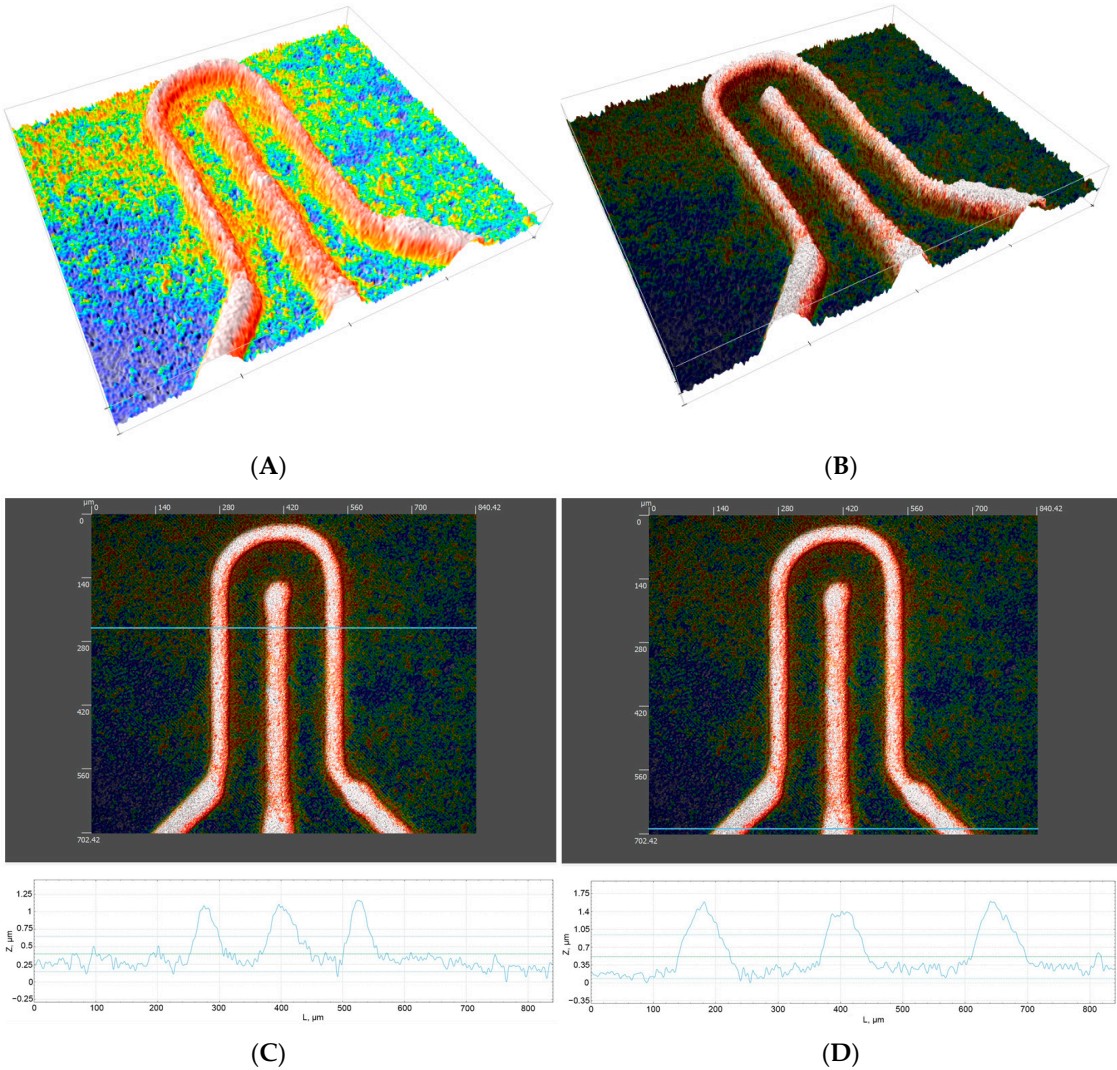

**Figure 12.** Images of a typical microheater annealed at 750 °C, which were rendered based on data acquired with a non-contact optical 3D profiler. (**A**,**B**) Perspective view in two color schemes. (**C**) Top view and profile of the narrowest part of the microheater. (**D**) Top view and profile of the initial part of the arms adjacent to the narrowest part. The presented profiles correspond to the positions of the light blue lines.

The microheaters were wired to the copper laminated glass epoxy substrates with 80 μm thick copper wires by attaching them to the terminal parts of the microheaters (see Figure 1) with PELCO colloidal silver (product No. 16031, Ted Pella, Inc., Redding, CA, USA). The wired microheaters were fed by a direct current (DC) from the power supply in constant voltage mode. To obtain the power dependence of the temperature of the hot part of the microheater, a series of images of its thermal radiation were acquired at various values of consumed power. Figure 14A demonstrates a typical thermal radiation image of the microheater, operated at a power of 200 mW. For comparison, Figure 14B presents an image of the same microheater operated at the same power taken on a Nikon Coolpix L820 digital camera. The temperature of the microheater at a given point could be retrieved from the signal of the corresponding image sensor by using the calibration coefficient. The calibration was performed on a small piece ($2 \times 2$ mm$^2$) of sintered material exfoliated from the substrate, which was transferred to the large thick film heater with high temperature uniformity. The temperature of the transferred piece of sintered material during the calibration was controlled with a thermocouple. By this means, the emissivity of the microheater was not required in the calculations, since the calibration was performed on the material constituting a microheater.

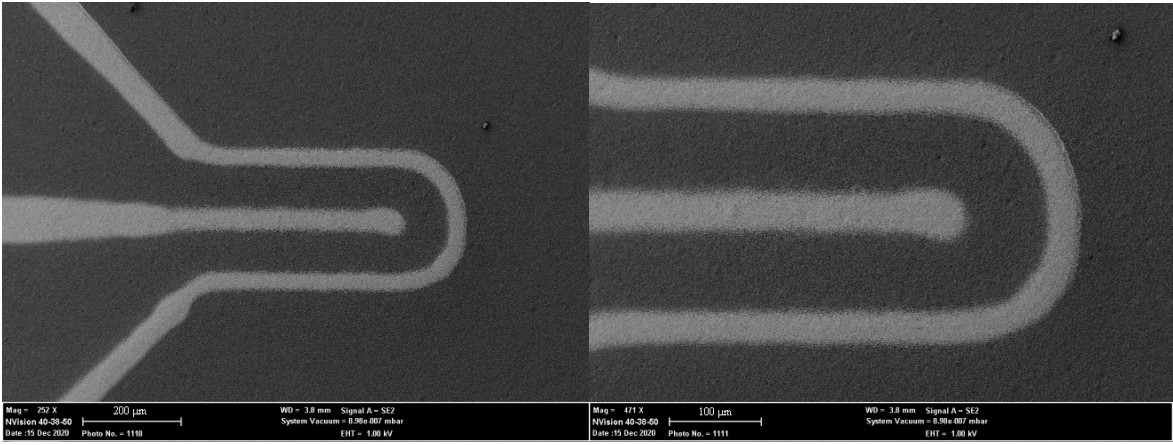

**Figure 13.** SEM images of the microheater presented in Figure 12.

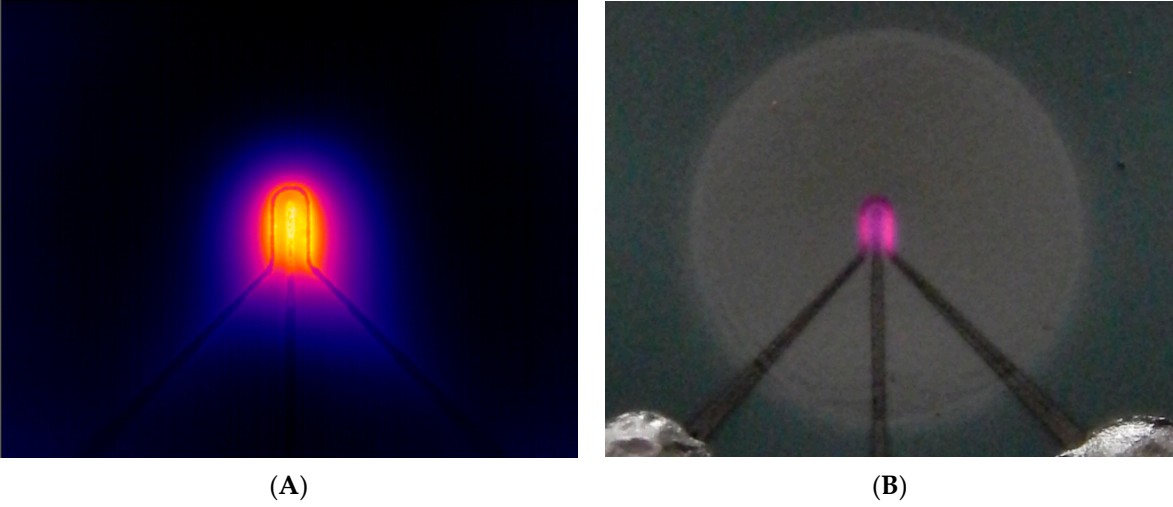

(**A**)                                                    (**B**)

**Figure 14.** Thermal radiation image of the hot part of the microheater operating at a power of 200 mW, acquired with a FLIR SC655 thermal imaging camera (**A**), and an image of the same microheater operating at the same power, taken with a Nikon Coolpix L820 digital camera (**B**).

Figure 15 presents the typical power dependence of the temperature of the hot part of the microheater. At a maximum temperature required for the operation of the MOS gas sensors of 500 °C, the power consumption was about 140 mW, being comparable with that of microheaters used in commercial MEMS-based MOS gas sensors [38]. The consumed power could be readily reduced by a factor of about 1.5 by printing each arm of the microheater with a wider angle (up to 30 degrees) between two sides. This would result in a sharper decrease in the temperature of the microheater arms in the direction of the boundary of the hole in the LTCC wafer, due to the sharper decrease in the current density.

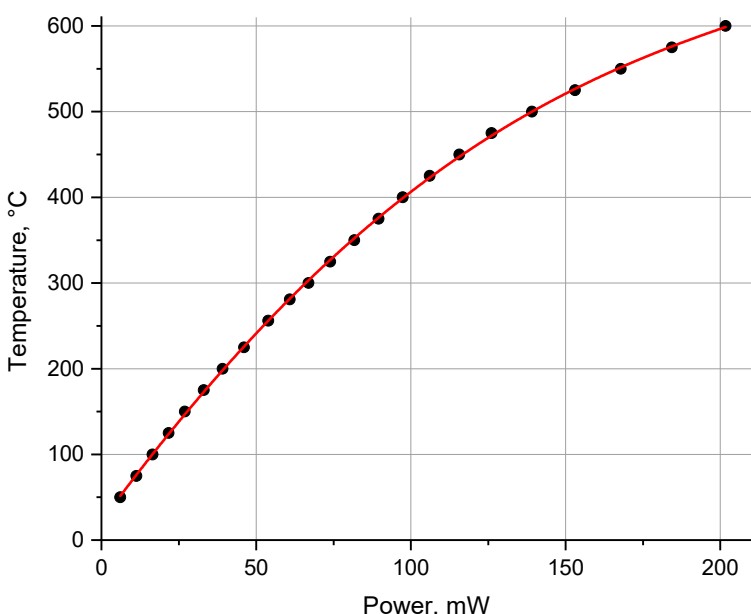

**Figure 15.** Power dependence of the temperature of the hot part of the microheater.

### 3.3. Catalytic Activity of Synthesized Nanoparticles

To demonstrate the catalytic activity of the synthesized nanoparticles, we measured the thermal responses of the non-sintered microheaters (heat treated at 400 °C) to hydrogen mixed with dry air. The heat treatment of printed ink at 400 °C led to the formation of a binder-free array of nanoparticles with a high specific surface, thus making it possible to detect changes in the microheater resistance caused by exothermic reactions occurring on the surface of the nanoparticles, to which catalytic hydrogen combustion [39] and a reduction of the remaining platinum oxide by hydrogen [40] could be referred. Therefore, the non-sintered microheater serves the functions of both a heating element and a catalytic layer. Figure 16 presents typical resistance transients upon exposure to 2500 ppm of hydrogen, which were retrieved from the current transients measured in constant voltage mode. The measurements were carried out at a voltage of 2 V and 3 V. The average power consumption of the microheater was estimated to be 45 mW (2 V) and 100 mW (3 V). These values corresponded to a temperature of the hot part of about 220 °C (2 V) and 410 °C (3 V), respectively. The drift of the microheater resistance during exposure to hydrogen can be explained by a gradual sintering of the metallic phase due to the temperature growth caused by the exothermic reactions mentioned above. Further possible applications of spark discharge-synthesized nanoparticles as a catalyst (e.g., in pellistors) should involve the stage of annealing the deposited ink at temperatures above 600 °C in order to completely reduce the platinum oxide to a metallic state. To provide a high specific surface for a catalytic layer, it has to be formed within a high-porosity ceramic carrier made of alumina, in the case of pellistors [41].

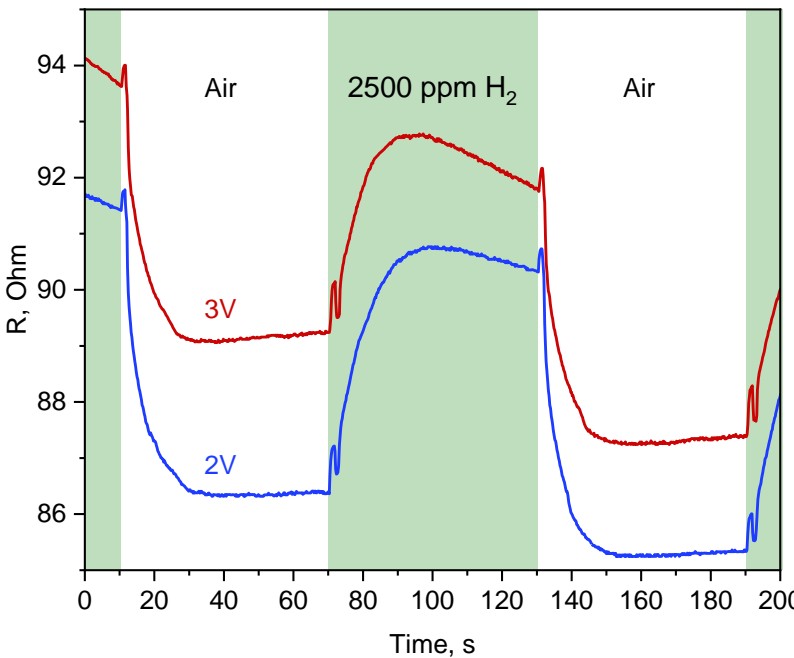

**Figure 16.** Resistance transients of a non-sintered microheater upon exposure to 2500 ppm of hydrogen mixed with air obtained at a voltage of 2 V (blue curve) and 3 V (red curve).

## 4. Conclusions

Spark ablation technology was applied for producing nanoparticles from platinum ingots as a feed material by using air as a carrier gas. The synthesized nanomaterial, composed of an amorphous platinum oxide PtO (83 wt. %) and crystalline metallic platinum (17 wt. %), was used for formulating functional colloidal ink. The formulated ink was characterized by the following principal parameters: a concentration of platinum of 24.5 wt. %, viscosity of 11.4 mPa·s (25 °C), surface tension of 43.9 mN/m (25 °C) and boiling point of the solvent of 119.6 °C. Annealing of the deposited ink at 750 °C resulted in the formation of a polycrystalline material with a resistivity of $1.2 \cdot 10^{-7}$ $\Omega$·m (25 °C) comprising 99.7 wt. % of platinum. To demonstrate the possibility of the application of formulated ink in printed electronics, we patterned conductive lines and microheaters on alumina substrates and 20 μm thick LTCC membranes with the use of aerosol jet printing technology. The power consumption of the microheaters fabricated on the LTCC membranes was found to be about 140 mW at a temperature of the hot part of 500 °C, thus allowing one to consider these structures as promising micro-hotplates for MOS gas sensors. The catalytic activity of the synthesized nanoparticles was demonstrated by measuring the thermal responses of the non-sintered microheaters (heat treated at 400 °C) to hydrogen, which opens the prospective application of spark discharge-produced platinum-based powders in catalysis, in particular for the fabrication of catalytic pellistors.

**Author Contributions:** Conceptualization, I.A.V. and A.A.V.; methodology, I.A.V., A.A.E. and V.V.I.; investigation, I.A.V., I.S.V., N.P.S., T.L.S., A.S.M., V.I.B., P.V.A., Y.Y.L., A.M.M., A.A.L., V.A.B. and A.E.V.; resources, I.A.V., A.A.V., Z.L., E.P.S. and V.V.I.; data curation, I.A.V., N.P.S., T.L.S. and I.S.V.; writing—original draft preparation, I.A.V., I.S.V. and A.A.V.; writing—review and editing, I.A.V., I.S.V., N.P.S. and T.L.S.; visualization, I.A.V., I.S.V., N.P.S. and T.L.S.; supervision, I.A.V. and A.A.V.; project administration, I.A.V., A.A.E. and V.V.I.; funding acquisition, I.A.V. and V.V.I. All authors have read and agreed to the published version of the manuscript.

**Funding:** This research was funded by the Ministry of Science and Higher Education of the Russian Federation (state contract no. 075-00337-20-03, project identifier FSMG-2020-0007, project title: Development of functional materials with controlled electrical, chemoresistive and catalytic properties for manufacturing sensor microsystems by using methods of printed electronics).

**Institutional Review Board Statement:** Not applicable.

**Informed Consent Statement:** Not applicable.

**Data Availability Statement:** The data presented in this study are available in this article.

**Conflicts of Interest:** The authors declare no conflict of interest.

### Appendix A

**Table A1.** Results of the characterization of the powder produced by spark discharge and the sintered material.

| | Powder | Sintered Material |
|---|---|---|
| Elemental composition (ICP-AES) | Pt: 92.7 wt. % <br> Sn: 0.077 wt. % <br> Cu: 0.074 wt. % <br> Ag: 0.033 wt. % <br> Pd: 0.022 wt. % <br> Fe: 0.014 wt. % <br> Zn: 0.004 wt. % <br> Ca: 0.004 wt. % | Pt: 99.7 wt. % <br> Sn: 0.086 wt. % <br> Cu: 0.072 wt. % <br> Ag: 0.037 wt. % <br> Pd: 0.023 wt. % <br> Fe: 0.020 wt. % <br> Zn: 0.014 wt. % <br> Ca: 0.005 wt. % |
| Phase composition (XRD, XPS) | (1) Crystalline metallic platinum (Fm3m crystal structure) = 17 wt. %. <br> (2) Amorphous phase (PtO) = 83 wt. % | Crystalline metallic platinum (Fm3m crystal structure)–more than 99 wt. %. |
| Particle size parameters determined from TEM images Crystalline phase identified from SAED patterns | (1) Number-weighted modal size = 13 nm; volume-weighted modal size = 25 nm. <br> (2) The crystalline phase is represented by platinum with the Fm3m crystal structure. | – |
| Grain size (SEM) | – | 100–400 nm |
| Resistivity | – | $1.2 \cdot 10^{-7}$ $\Omega \cdot$m (1.1 times greater than that of the bulk Pt due to the insignificant residual porosity) |

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
