# Peer review of "Platinum Based Nanoparticles Produced by a Pulsed Spark Discharge as a Promising Material for Gas Sensors"

_applsci, doi:10.3390/app11020526_

Round 1
Reviewer 1 Report
It is an interesitng manuscript about an application of a spark ablation technology for producing nanoparticles from platinum. The synthesized nanomaterial composed of the mixture of amorphous platinum oxide and metallic platinum was used for preparation of a colloidal ink. The following annealing of the deposited ink results in the reduction of PtO to metallic platinum. It was also demonstrated the possibility of application of formulated ink in printed electronics, in particularly, in an use of aerosol jet printing technology. It was shown that micro-hotplates prepared usig this technology could be applied for MOS gas sensors.
The manuscript could de accepted for a publcation in Advanced Science with minor revisions (s. comments below).
Comment 1. The use of an air as carrier gas is rather questionable, because it is difficult to control its purity, humidity, etc. All these parameter could effect of formation of PtO/Pt nanoparticles. I believe the use of syntetic gas (N2/O2 mixture) or O2 could be more effective.
Comment 2. Line 229 and Figure2: Here authors provided an information about the Pt4f7/2 line at 74 eV, but I didn't find any peaks at ~74/78 eV in the experimental data/fitting.
The strong assymetry of Pt4f(0) peaks supposes the presence of Pt(2+) and Pt(4+) in the sintered sample, but these phases are not shown in the fitting data. I ask Author to improve Figure 2 correspondingly.
Comment 3: Figure 3: I ask Authors to provide an assignment of diffraction peaks (reflections).
Comment 4. Figure 5: I recommend to use the same X-scale for Figure 5A and 5B.
Comment 5: Lines 317-318: Please correct these lines in an accordance with Comment 2 (the second part).
Reviewer 2 Report
The reviewed manuscript presents interesting topic - this is fabrication of platinum-based nanoparticles by pulsed spark discharges. Just after spark discharges Authors received mixture of Pt and PtO nanoparticles but after subsequent thermal processes they obtained almost pure platinum nanoparticles. Such nanoparticles were used for fabrication of colloidal ink which were used for fabrication of microheaters by aerosol jet printing. The structure of partially sintered microheaters were used as possible pellistor structure exhibiting sensitivity to hydrogen.
The paper is very interesting and well prepared. I found only a few weak points listed below:
- There is a contradiction in the information contained in lines 334-336 and in the caption to Fig. 10. Which information is true?
- Comparing the behaviour of structures of microheaters fired at 750 deg C and 400 deg C, one should show their resistance at room temperature and their temperature coefficient of resistance. Probably on this basis one could conclude about the relationship between Pt and PtO in such films depending on their temperature treatment.
- I would like to point out that for such narrow and thin layers it is difficult to obtain a constant cross-section (please see eg. M. Gierczak, P. Markowski, Z. Zaluk, A. Dziedzic, P. Jankowski-Mihulowicz; Ink-jet printed conductive films – geometrical and electrical characterization, Proc. 39th Int. Spring Seminar on Electronics Technology, Pilsen (Czechia), May 2016, s.392-397; DOI: 10.1109/ISSE.2016.7563227). What is the situation like in the case of microheaters?
Based on above remarks I propose to accept this manuscript after minor revision.
